# Transcriptional Regulation of Protein Phosphatase 2C Genes to Modulate Abscisic Acid Signaling

**DOI:** 10.3390/ijms21249517

**Published:** 2020-12-14

**Authors:** Choonkyun Jung, Nguyen Hoai Nguyen, Jong-Joo Cheong

**Affiliations:** 1Department of International Agricultural Technology and Crop Biotechnology, Institute/Green Bio Science and Technology, Seoul National University, Pyeongchang 25354, Korea; jasmin@snu.ac.kr; 2Department of Plant Science, College of Agriculture and Life Sciences, Seoul National University, Seoul 08826, Korea; 3Faculty of Biotechnology, Ho Chi Minh City Open University, Ho Chi Minh City 700000, Vietnam; nguyen.nhoai@ou.edu.vn; 4Center for Food and Bioconvergence, Seoul National University, Seoul 08826, Korea

**Keywords:** abscisic acid, chromatin remodeling, drought, guard cell, osmotic stress, protein phosphatase 2C, salinity, stomata, stress memory, transgenerational inheritance

## Abstract

The plant hormone abscisic acid (ABA) triggers cellular tolerance responses to osmotic stress caused by drought and salinity. ABA controls the turgor pressure of guard cells in the plant epidermis, leading to stomatal closure to minimize water loss. However, stomatal apertures open to uptake CO_2_ for photosynthesis even under stress conditions. ABA modulates its signaling pathway via negative feedback regulation to maintain plant homeostasis. In the nuclei of guard cells, the clade A type 2C protein phosphatases (PP2Cs) counteract SnRK2 kinases by physical interaction, and thereby inhibit activation of the transcription factors that mediate ABA-responsive gene expression. Under osmotic stress conditions, PP2Cs bind to soluble ABA receptors to capture ABA and release active SnRK2s. Thus, PP2Cs function as a switch at the center of the ABA signaling network. ABA induces the expression of genes encoding repressors or activators of PP2C gene transcription. These regulators mediate the conversion of PP2C chromatins from a repressive to an active state for gene transcription. The stress-induced chromatin remodeling states of ABA-responsive genes could be memorized and transmitted to plant progeny; i.e., transgenerational epigenetic inheritance. This review focuses on the mechanism by which PP2C gene transcription modulates ABA signaling.

## 1. Introduction

The current global climate crisis has resulted in long spells of dry weather and a shortage of rainfall, and becomes a serious threat to crop productivity and food supply. Under drought conditions, the salt concentration increases as the moisture content decreases in the soil. Water deficit and salinity inflict osmotic stress on plant cells. Plants are not able to escape from adverse environments, and so respond to such stressful conditions by triggering physiological and cellular responses [1,2,3]. Most prominently, plants close stomatal apertures on the epidermis to limit transpiration and thereby prevent loss of water under osmotic stress conditions. A stomatal aperture is formed by two flanking guard cells that swell or deflate by regulating turgor pressure through ionic fluxes via ion channels anchored in the plasma membrane [4].

Under osmotic stress conditions, plants biosynthesize and accumulate abscisic acid (ABA), a sesquiterpenoid hormone [5]. Most importantly, ABA functions as a chemical messenger that induces numerous genes whose products are crucial for stomatal closure and the accumulation of osmoprotectants [6,7,8]. A previous transcriptomic study showed that more than half of the genes regulated by ABA treatment are also induced under drought or salinity conditions [9]. Likewise, ABA deficiency impairs osmotic stress regulation of gene expression [10]. Thus, it appears that osmotic stress-induced expression of the responsive genes is entirely dependent on ABA. Because plants encounter not only osmotic stress but also abnormal temperatures (heat and cold) and biotic stresses (pathogens and insects) in nature, ABA signaling is integrated with other ABA-independent signaling pathways [11,12].

ABA is mainly biosynthesized in vascular tissues and transported to sites of action, such as guard cells [13,14]. In guard cells, ABA molecules are perceived by receptors in the nucleus and cytosol, activating the sucrose non-fermenting 1-related protein kinase 2 (SnRK2) family of protein kinases [15,16]. In the nucleus, SnRK2s phosphorylate a number of transcription factors that activate transcription of the ABA-responsive genes whose products are implicated in stress responses and tolerance. Inversely, the clade A type 2C protein phosphatases (PP2Cs) counteract SnRK2s by physical interaction, exerting negative regulation of ABA signaling [17]. Under osmotic stress conditions, PP2Cs bind to ABA receptors to capture ABA, releasing and activating the SnRK2s. Thus, PP2Cs function as a switch at the center of the ABA signaling network.

In Arabidopsis, nine protein phosphatases are classified as clade A PP2Cs [18,19,20]. Six of them—ABA insensitive 1 (ABI1), ABI2, ABA hypersensitive germination 1 (AHG1), AHG3/PP2CA, hypersensitive to ABA1 (HAB1), and HAB2—are involved in ABA signaling in the osmotic stress response. The remaining three members, highly ABA-induced 1 (HAI1), PP2C1/HAI2, and HAI3, affected ABA-independent low water potential phenotypes, such as enhanced accumulation of osmoprotectants and suppression of the expression of abiotic stress-associated genes encoding dehydrins and late embryogenesis abundant proteins (LEAs) [21]. ABI1 and ABI2 are main components of ABA signaling under abiotic stresses and in developmental processes [22,23]. The dominant ABA response mutants of Arabidopsis, *abi1* and *abi2*, were originally isolated on the basis of their ABA insensitivity reflected in reduced seed dormancy and in symptoms of withering [24]. However, it was subsequently found that all of the knockout mutants of PP2C genes exhibited significant ABA hypersensitivity, indicating that they are negative regulators of ABA signaling. Recessive *hab1-1* mutants also showed enhanced ABA-responsive gene expression, increased ABA-mediated stomatal closure, and ABA-hypersensitivity in seed germination, indicating that HAB1 also negatively regulates ABA signaling [25,26].

ABA also plays pivotal roles in various physiological processes during the plant life cycle, including seed dormancy, germination, lateral root formation, light signaling convergence, and control of flowering time [5,7,12]. These functions of ABA are related to Ca^2+^ influx, the production of reactive oxygen species such as H_2_O_2_, ion transport, and electrical signaling [11,12,27]. During these processes, ABA signaling interacts antagonistically or synergistically with other hormonal signaling pathways mediated by auxin, cytokinin, ethylene, and jasmonates [7]. Thus, excess ABA impairs developmental processes such as senescence, as well as pollen fertility, and also leads to seed dormancy and susceptibility to diseases [28].

Stomata control transpiration and CO_2_ uptake by optimizing the aperture size in response to various environmental and endogenous signals, including ABA, light, and CO_2_ [29,30,31,32]. ABA causes stomatal closure, but light induces the opening of stomata to enhance CO_2_ assimilation for photosynthesis. Plants often integrate osmotic stress and light signals simultaneously, and so the stomatal pores are opened and closed to maintain homeostasis.

Plants finely control the ABA concentration and ABA signaling during and after exposure to stressful conditions. The ABA levels in tissues are controlled by biosynthesis and catabolism [5]. In addition, the ABA signaling network can be desensitized by degradation of core proteins by the ubiquitin proteasome system [33]. In addition, plant cells modulate the ABA signaling pathway via PP2C-madiated negative feedback regulation.

ABA regulates the PP2C concentration by inducing the expression of genes encoding transcriptional repressors or activators. These transcriptional regulators compete with the PP2C gene promoters, inducing chromatin remodeling and thus the switch from a repressive to an active state. In this manner, ABA simultaneously activates positive and negative regulatory systems affecting its own signaling pathway. The chromatin state acquired for osmotic stress tolerance can be memorized and transmitted to newly developed cells during vegetative growth [34,35] and even inherited by the next generation of plants; i.e., transgenerational epigenetic inheritance [36,37].

In this article, we reviewed how plants modulate the ABA signaling pathway, focusing on the transcriptional regulation of PP2C gene expression by ABA. The biosynthesis, signaling mechanisms, and biological functions of ABA were recently reviewed comprehensively [38,39]. The epigenetic regulation of plant responses to abiotic stresses, including ABA treatment, drought, and salinity, were also reviewed in detail [40,41,42]. Kumar et al. [12] reviewed the integration of ABA signaling with other signaling pathways in development and plant stress responses.

## 2. Roles of PP2Cs in ABA Signaling

### 2.1. Negative Regulation of ABA Signaling

High levels of PP2Cs are part of the negative feedback mechanism that desensitizes plants to high ABA levels [43,44]. In the absence of ABA, PP2Cs physically interact with SnRK2s to form complexes (Figure 1A). In Arabidopsis, subgroup III SnRK2s are key regulators of ABA signaling [45,46]. There are 10 SnRK2 members in Arabidopsis; i.e., SnRK2.1–SnRK2.10. Among them, SnRK2.2, SnRK2.3, and SnRK2.6/OST1 are the strongest activators of ABA responses, and so are regarded as primary regulators of ABA signaling. The triple mutation (*snrk2.2/2.3/2.6*) largely blocked the major ABA responses [47]. ABI1 interacts with SnRK2.6/OST1, SnRK2.2, and SnRK2.3 in plants, resulting in the inactivation of downstream components; e.g., AREB/ABFs transcription factors and ion channels [46].

The SnRK2.6/OST1 was characterized as a critical limiting component in ABA regulation of stomatal apertures, ion channels, and NADPH oxidases in Arabidopsis guard cells [48]. PP2Cs dephosphorylate Ser175 in the activation loop of SnRK2.6, resulting in deactivation of the kinase [17]. Several PP2C-interacting factors, such as enhancer of ABA coreceptor 1 (EAR1) and PR5-like receptor kinase 2 (AtPR5K2), enhance the phosphatase activity of PP2Cs by phosphorylating them, and so modulate plant responses to drought stress [49,50].

### 2.2. Perception of ABA Signal

ABA molecules biosynthesized in vascular tissues are distantly transmitted to sites such as guard cells to activate the closure of stomata [13,14]. Multiple ABA transporters have been identified in Arabidopsis, including exporters (AtABCG25 and AtDTX50) and importers (AtABCG40 and AtAIT1) [51,52,53,54,55]. Guard cells themselves also biosynthesize ABA, which is sufficient for stomatal closure in response to low air humidity [56].

ABA molecules are perceived intracellularly by soluble receptors predominantly located in the nucleus and cytosol of guard cells [16,57]. A number of synonymous ABA receptors, e.g., pyrabactin resistance (PYR), PYR-related (PYL), and regulatory component of the ABA receptor (RCAR), have been identified as PP2C-interacting proteins in Arabidopsis [58,59,60]. PP2Cs have direct physical interactions with ABA and ABA receptors; these interactions are required for high-affinity binding of ABA [61,62]. Each PP2C functions as an ABA co-receptor within a holoreceptor complex that is constructed in combination with a particular PYR/PYL/RCAR.

The Arabidopsis genome contains 14 PYR/PYL/RCAR genes, which encode small proteins containing highly conserved amino acid residues [63]. All of them (except *PYL13*) are able to activate ABA-responsive gene expression. Transgenic lines expressing nuclear *PYR1* in an ABA-insensitive mutant background exhibited ABA responses, but cytosolic PYR1 was also required for full recovery of ABA responses [64]. PYL8/RCAR3 showed subcellular localization mainly in the cytosol and nucleus, and its overexpression led to enhanced drought resistance of Arabidopsis [65]. Guard cells express the six ABA receptor genes *PYR1*, *PYL1*, *PYL2*, *PYL4*, *PYL5*, and *PYL8* to mediate stomatal closure [66,67]. Arabidopsis mutants lacking three, four, five, and six of these PYR/PYL/RCAR genes (*pyr1/pyl1/pyl2/pyl4/pyl5/pyl8*) exhibited gradually increased stomatal conductance, indicating that this family of receptors quantitatively regulates the stomatal aperture [66]. Dittrich et al. [67] proposed that response specificity is achieved when the signals stimulate different members of the PYR/PYL/RCAR receptor family; PYL2 is sufficient for ABA-induced guard cell responses, whereas PYL4 and PYL5 are essential for the responses to CO_2_. Different combinations of PYRs and PP2Cs influence ABA binding affinity, and therefore affect the ABA sensitivity of the whole plant [68,69].

ABA directly binds to the PYR/PYL/RCAR proteins [61,62,70,71]. ABA binding leads to conformational changes of the ABA receptors, which allows physical interaction with PP2Cs and inhibits phosphatase activity [72,73,74] (Figure 1B). Nishimura et al. [74] performed co-immunoprecipitation experiments in a transgenic Arabidopsis line stably transformed with yellow fluorescent protein (*YFP*)–*ABI1* fusion genes using a PYR1 antibody, and observed that the ABI1–PYR1 interaction was induced within 5 min after exogenous ABA application. Remarkable similarity was found in PP2C recognition between SnRK2 and ABA receptors [75,76]. In the absence of ABA, PP2C binds to the SnRK2 kinase domain and dephosphorylates Ser 175 in the activation loop. Upon perception of ABA, ABA receptor binds to PP2C by inserting the gate loop into the PP2C active cleft.

### 2.3. Regulation of ABA-Responsive Gene Expression

Upon the formation of PYL-ABA-PP2C complexes, SnRK2s dissociate from inactivated PP2Cs and recover their kinase activity. ABA treatment and osmotic stress stimulate phosphorylation of Ser 175 in the activation loop of SnRK2.6 [77]. When released from PP2C inhibition, SnRK2.6 autophosphorylates at Ser175 and Thr176 to recover full activity [76]. Free and active SnRK2s subsequently phosphorylate and activate downstream transcription factors in the nucleus and ion channels in the cytosol [57].

In the nucleus, the SnRK2-mediated phosphorylation of transcription factors results in the expression of numerous ABA-responsive genes. By analyzing the promoters of ABA-responsive genes, a conserved ABA-responsive element (ABRE; PyACGTGG/TC) was identified [78,79]. Subsequently, a number of ABRE-binding (AREB) proteins and ABRE-binding factors (ABFs) were identified by yeast one-hybrid screenings [80,81]. AREB/ABFs belong to the basic-domain leucine zipper (bZIP) transcription factor family and are colocalized with SnRK2s in plant cell nuclei [46]. Multiple conserved RxxS/T sites in AREB/ABFs are phosphorylated in an ABA-dependent manner [81,82,83].

Among the nine AREB/ABFs in Arabidopsis, ABF1, AREB1/ABF2, ABF3, and AREB2/ABF4 act as master transcription factors in ABA signaling for osmotic stress tolerance [84]. Overexpression of these genes in Arabidopsis resulted in ABA hypersensitivity and enhanced drought stress tolerance [85,86,87]. By contrast, the triple knockout mutant (*areb1/areb2/abf3*) displayed impaired expression of ABA- and osmotic stress-responsive genes, resulting in increased sensitivity to drought [88]. Fujii et al. [89] reconstituted ABA-triggered phosphorylation of ABF2/AREB1 in vitro by combining PYR1, ABI1, and SnRK2.6/OST1, demonstrating that PYR/PYL/RCAR receptors, PP2Cs, and SnRK2s constitute the core of the ABA signaling pathway.

## 3. Transcriptional Regulation of PP2C Gene Expression

### 3.1. ABA-Induced PP2C Gene Expression

ABA induces the expression of AREB/ABF genes, resulting in the accumulation of endogenous AREB/ABF proteins [6]. Concurrently, expression of the group-A PP2C genes is highly inducible in response to ABA and abiotic stresses [6,20]. The induction of PP2C gene expression may be an ABA desensitization mechanism modulating ABA signaling and maintaining plant homeostasis. Therefore, ABA upregulates genes encoding both positive and negative effectors of its signaling network.

The ABA-induced expression of PP2C genes is also mediated by AREB/ABFs. In response to salt stress, the transcript levels of PP2C genes (*ABI1*, *ABI2*, and *HAI1*) in an *abf3* mutant were markedly lower than those in wild-type plants [90], supporting a positive role for ABF3 in the activation of PP2C genes. A number of ABF3-binding sites, TCACGttt and ACACGgtt [91], are present in the promoter regions of these PP2C genes. In fact, a transcription factor hierarchy showed that ABF3 directly associates with the promoters of these genes [92]. Furthermore, Wang et al. [93] demonstrated that ABF transcription factors (i.e., ABF1 to ABF4) directly bind to the promoters of PP2C genes (*ABI1* and *ABI2*), and mediate rapid induction of their expression upon exogenous ABA treatment.

These data indicate that ABFs mediate ABA-induced expression of PP2C genes, thus playing a role in the negative feedback regulation of ABA signaling, in addition to the ABA-induced expression of ABA-responsive genes. Therefore, ABFs play dual in both the forward and backward regulation of ABA signaling. The ABF-mediated transcriptional upregulation of PP2Cs and PP2C-mediated inactivation of ABFs constitute a tight regulatory loop in ABA signaling modulation.

### 3.2. Repression of PP2C Gene Transcription

Under normal conditions, the expression of PP2C genes is maintained at basal levels, while under osmotically stressful conditions, the expression of PP2C genes is suppressed to enhance ABA signaling. A couple of MYB transcription factors were reported to act as repressors of PP2C gene transcription. For instance, *AtMYB44* transcripts accumulated under ABA treatment and abiotic stresses such as dehydration, low temperature, and salinity [94,95,96]. Microarray and northern blot analyses revealed that salt-induced expression of a group of PP2C genes, including *ABI1*, *ABI2*, *AtPP2CA*, *HAB1*, and *HAB2*, was significantly repressed in transgenic Arabidopsis overexpressing *AtMYB44* [94,95]. The transgenic plants showed increased sensitivity to ABA and more rapid ABA-induced stomatal closure. Under drought conditions, the transgenic Arabidopsis exhibited reduced rates of water loss and enhanced tolerance [94]. Furthermore, transgenic soybean [97] and rice seedlings [98] overexpressing *AtMYB44* exhibited significantly enhanced drought and salt stress tolerance. It appears that the enhanced osmotic stress tolerance of the transgenic plants was conferred by reduced expression of genes encoding PP2Cs that function as negative regulators of ABA-mediated stomatal closure. Cui et al. [99] also showed that the expression of a group of PP2C genes, such as *ABI1*, *ABI2*, and *PP2CA*, was suppressed in *AtMYB20*-overexpressing transgenic lines, but induced in *AtMYB20*-repression lines in response to salt treatment.

A number of the AtMYB44-binding sequences of AACnG [100] exist in transcription start site (TSS) regions of *ABI1*, *ABI2*, and *HAI1*. A chromatin immunoprecipitation (ChIP) assay demonstrated that AtMYB44 binds to the promoters of these genes under normal conditions to repress gene transcription [101]. In response to salt stress, AtMYB44 binding to PP2C promoters was significantly reduced, and the transcript levels of the genes were increased [90]. These results confirmed that AtMYB44 acts as a repressor of PP2C gene transcription. Such promoter-binding and repressive functions of AtMYB44 were also observed for *AtMYB44* [102] and *AtLEA4-5* [103].

A number of independent studies suggested that AtMYB44 physically interacts with ABA receptors. Jaradat et al. [96] observed that AtMYB44 (synonym MYBR1) physically interacts with PYL8 and represses ABA signaling in response to drought and senescence. Binding to PYL8 may block the interaction of AtMYB44 with PP2Cs or promoter of ABA-responsive genes. Li et al. [104] showed that AtMYB44 and ABI1 competed for binding to PYL9 and thereby reduced the inhibitory effect of the receptor on ABI1 phosphatase activity in the presence of ABA. These results suggest that AtMYB44 may act as a negative regulator of ABA signaling, which is inconsistent with its reported indirect positive role of suppressing PP2C gene transcription. Further studies are needed to explore the role of AtMYB44 as a positive or negative (or dual) regulator of ABA signaling.

## 4. Epigenetic Regulation of ABA Signaling

### 4.1. Epigenetic Regulation of ABA-Responsive Gene Transcription

In the chromatin of eukaryotic cells, genomic DNA is wrapped around a histone octamer consisting of H2A, H2B, H3, and H4 to form a nucleosome [105]. The access of RNA polymerase to the chromatin is regulated by competition between transcription factors and nucleosomes [106,107,108]. Thus, the chromatin around the gene transitions from a repressive state into an active state to enable access by RNA polymerase [109]. Chromatin remodeling is accompanied by histone modification (acetylation and methylation), DNA methylation, and microRNA generation, which take place mainly in the promoter region close to TSS [110,111]. Activators loaded on the promoter recruit co-activators and histone acetyltransferases (HATs) that acetylate the histones and relax DNA–histone binding in chromatin [112]. Inversely, repressors recruit corepressors associated with histone deacetylases (HDAs) so that nucleosomes bind tightly to DNA.

Epigenetic chromatin modification plays an important role in plant responses to osmotic stress [113,114,115]. Histone acetylation is involved in the transcriptional regulation of genes encoding PP2C family proteins, such as ABI1 and ABI2 [116]. Conversely, a histone deacetylation complex targets the promoters of the genes encoding PYL4, PYL5, and PYL6, thereby repressing gene expression [117]. Ryu et al. [118] reported that a histone deacetylation complex containing HDA19 binds to the promoter region of ABI3, and subsequently represses its expression. In addition, ABA enhances the methylation of promoter DNA, repressing the expression of ABA-repressive genes in Arabidopsis [119]. Moreover, ABA upregulates the expression of microRNAs in Arabidopsis, such as *miR159*, *miR393*, and *miR402* [120,121,122].

The switch/sucrose non-fermenting (SWI/SNF) chromatin remodeling complex regulates gene transcription in plants [123,124]. A subunit of the complex, BRAHMA (BRM), hydrolyzes ATP to supply the energy necessary to alter the interaction of nucleosomes with DNA, and thereby change the position and occupancy [125,126,127]. A whole-genome mapping and transcriptome analysis revealed that BRM complex occupies thousands of sites in the Arabidopsis genome, where it contributes to the activation or repression of gene transcription [128]. Han et al. [129] showed that the BRM complex in Arabidopsis represses ABA responses by affecting the stability of the associated nucleosome at a transcription factor (ABI5) locus, thus inactivating the gene. However, it is unclear how the BRM-containing SWI/SNF complexes access and occupy their target loci. Arabidopsis BRM contains several DNA-binding and nucleosome-binding regions, in addition to the AT-hook region [130]. In a study of vegetative development and flowering, BRM complex was recruited to specific loci by physical interaction with a plant-unique H3K27me3 demethylase that targets specific genomic loci [131].

Peirats-Llobet et al. [132] reported that SnRK2.2/2.3/2.6 kinases directly interacted with BRM, which led to phosphorylation and inhibition of its activity, while PP2CA-mediated dephosphorylation restored the ability of BRM to repress the ABA response. In this case, a phosphorylation-based switch mediated by SnRK2 and PP2C controls the BRM-associated chromatin remodeling state, thereby regulating the transcription of ABA-responsive genes.

### 4.2. Chromatin Remodeling for PP2C Gene Expression

AtMYB44 contains the amino acid sequence LxLxL, a putative ethylene-responsive element binding factor-associated amphiphilic repression (EAR) motif. Many studies have demonstrated physical interactions among EAR-containing repressors and TOPLESS (TPL) corepressor [133]. Ryu et al. [118] observed that the transcription factor BES1 forms a repressor complex with TPL and HDA19, directly facilitating the histone deacetylation of *ABI3* chromatin in Arabidopsis, although it remains unclear whether TPL–HDA19 interaction is direct or facilitated by adapter proteins. TPL-related (TPR) corepressors also recruit histone deacetylases such as HDA6 or HDA19, which are involved in various signaling pathways [134,135,136].

ChIP assay with transgenic Arabidopsis overexpressing the *AtMYB44-GFP* (green fluorescence protein) fusion gene revealed that AtMYB44 bound to PP2C gene (*ABI1*, *ABI2*, and *HAI1*) promoters to suppress gene transcription in a signal-independent manner [101]. Yeast two-hybrid and bimolecular fluorescence complementation (BiFC) assays demonstrated that AtMYB44 physically interacts with TPR1 and TPR3 corepressors through the EAR motif. Levels of histone H3 acetylation around the promoter and TSS proximal regions of *ABI1*, *ABI2*, and *HAI1* were markedly lower in *AtMYB44-*overexpressing transgenic plants than in wild-type plants. These results suggest that AtMYB44 forms a complex with TPR corepressors and recruits HDAs to suppress PP2C gene transcription (Figure 2A). Another repressor of PP2C gene transcription, AtMYB20 [99], also contains an EAR motif in the C-terminal side of the catalytic domain.

In response to salt stress, the AtMYB44 repressor was released and DNA–histone binding in nucleosomes were relaxed from the promoter regions [90], forcing chromatins to adopt an open structure (Figure 2B). Under these conditions, histone H3 acetylation (H3ac) around the TSS regions significantly increased. Wang et al. [93] demonstrated that ABFs bind to the promoters of PP2C genes and induce their transcription. Indeed, the salt-induced increases in PP2C gene (*ABI1*, *ABI2*, and *HAI1*) transcription were reduced in *abf3* plants [90]. In addition, whole Arabidopsis genome mapping revealed that BRM occupies, although does not directly bind to *ABI1* and *ABI2* gene promoters [128]. The Arabidopsis mutant *brm-3*, which shows moderately impaired BRM activity, produced more PP2C gene transcripts under salt stress conditions [90]. Thus, BRM contributes to the closed structure of PP2C chromatins, suppressing gene transcription.

### 4.3. Osmotic Stress Memory

A stressful condition enables plants to respond more promptly and strongly to repeated stress events [35,137]. For instance, Ding et al. [138] observed that Arabidopsis plants trained with previous dehydration events wilted much slower than non-trained plants under subsequent dehydration conditions. Virlouvet and Fromm [139] observed that the stomatal apertures in previously stressed Arabidopsis remain partially closed during a watered recovery period, facilitating reduced transpiration during subsequent dehydration stress. In addition, the rate-limiting ABA biosynthetic genes were expressed at much higher levels during watered recovery in the guard cells. Moreover, they performed a genetic analysis using mutants in the ABA signaling pathway, and found that SnRK2.2 and SnRK2.3 are important for stress memory of guard cells in the subsequent dehydration response.

In the memory responses, a subset of genes termed ‘memory genes’ are expressed at highly elevated or reduced levels during subsequent stress conditions. Numerous drought stress memory genes have been identified in Arabidopsis [140], maize [141,142], rice [143], potato [144], and soybean [145]. In Arabidopsis and soybean, drought-induced memory genes exhibiting elevated levels of transcripts include those involved in ABA-mediated tolerance responses to abiotic stresses, while the drought-repressed memory genes can be classified as light-harvesting- or photosynthesis-related genes [140,145]. When repeated dehydration stresses were imparted by air-drying, the Arabidopsis PP2C genes—including *ABI1, ABI2, HAB2, HAI2,* and *AtPP2CA*—did not exhibit expression patterns indicative of memory function [140]. By contrast, approximately 10 PP2C genes were identified as drought-induced memory genes in soybean grown in water-deprived soil [145]. The potential of stress memory to enhance crop productivity under drought conditions has been explored for a number of crops, including potato [146], wheat [147,148], and olive [149].

The most plausible mechanism underlying stress memory is changes in the chromatin architecture of memory gene loci [114,150,151]. For instance, histone methylation may act as a persistent epigenetic mark associated with transcriptional memory. H3K4me3 deposition in memory gene loci was higher than in non-memory genes after multiple exposures to drought stress [138,152]. Sani et al. [153] reported that hyperosmotic priming of Arabidopsis seedlings with transient mild salt treatment resulted in enhanced drought tolerance during a second stress exposure, leading to shortening and fractionation of H3K27me3 islands. Whatever the mechanism, such an epigenetically modified state may be transmitted mitotically to newly developed cells during the cell division process.

Furthermore, traits acquired under stressful conditions can be transmitted to progeny of the next generation [36,37,154]. Transgenerational epigenetic inheritance has been explored in crop breeding [155,156,157]. For instance, Raju et al. [158] developed an epigenetic breeding system in soybean for increased yield and stability, with RNAi suppression of a gene used to modulate developmental, defense, plant hormone, and abiotic stress response pathways. Verkest et al. [159] improved drought tolerance in canola by repeatedly selecting for increased drought tolerance in three generations. Tabassum et al. [160] observed that seed priming and transgenerational transmission improved tolerance to drought and salt stress in bread wheat. Zeng et al. [161] reported that multi-generation drought imposition mediated adaptation to drought condition in rice plants. Walter et al. [137] reported drought memory in grasses over an entire vegetation period, even after harvest and subsequent sprouting. However, net photosynthesis was reduced by 25% by recurrent drought treatment, which could have adverse effects on crop yield under more severe or longer droughts.

In general, the duration of a stress memory is relatively short, i.e., is limited to one generation [35,150,151]. Levels of the memory marker H3K4me3 in dehydration stress memory genes were elevated for 5 days [138]. This hampers application of stress memory to improve the stress tolerance of crops. In particular, although a number of studies have shown the involvement of epigenetic mechanisms, the principles underlying transgenerational inheritance are largely unknown [40,41]. Induced changes in the DNA methylation state were suggested as a possible mechanism by Zheng et al. [161], who observed that multi-generational drought stimulation induced the non-random appearance of epimutations and inheritance of high methylation state in advanced rice plant generations. As in animal cells, the acquired memory state could be reset (or forgot) during meiosis [162]. The mechanism by which plant cells overcome such resetting processes during meiosis and transmit the stress memory to progeny remains to be elucidated.

## 5. Conclusions and Perspectives

Drought and salinity are the most serious threats to crop productivity and food supply under global climate change. Therefore, understanding the mechanisms underlying osmotic stress tolerance and its application to crop breeding is an important topic in plant molecular science and biotechnology. ABA is a vital plant hormone that plays a key role in osmotic stress tolerance. ABA induces the closure of stomata in the epidermis, to limit transpiration and thereby prevent loss of water under osmotic stress conditions. The stomatal pores are open to uptake CO_2_ for photosynthesis, and thereby maintain plant homeostasis. Therefore, it is not always favorable to artificially enhance ABA biosynthesis and signaling by gene modification or editing. It is essential to gain insight into the strategies that plants use in nature to deal with adverse environments without any negative effects on development or growth. In plant guard cells, PP2Cs counteract SnRKs for negative feedback regulation of ABA-induced stomatal closure. ABA induces both positive and negative mechanisms that modulate ABA responses by regulating PP2C gene transcription. Finally, plants encounter not only osmotic stress, but also temperature and biotic stresses. Therefore, communication between signaling pathways under different combinations of stresses should be more intensely investigated. Understanding the molecular mechanisms underlying stress memory and transgenerational inheritance might provide new methods to breed higher-quality crops that can withstand adverse climatic conditions.

## Figures and Tables

**Figure 1 ijms-21-09517-f001:**
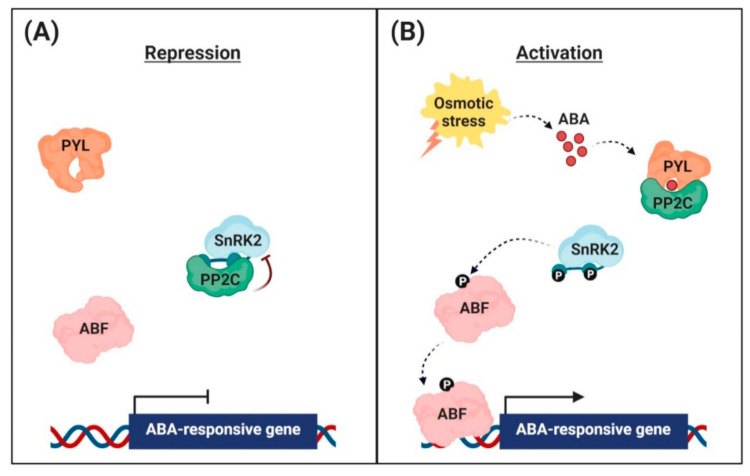
Abscisic acid (ABA) signaling pathway in the nuclei of guard cells. (**A**) Repression of ABA-responsive gene expression. In the absence, the clade A protein phosphatases (PP2Cs) physically interact with the sucrose non-fermenting 1-related protein kinases (SnRK2s) to reduce kinase activity via dephosphorylation. This inhibits the activity of ABRE-binding (AREB)/ABRE-binding factor (ABF) transcription factors and suppression of ABA-responsive gene transcription. (**B**) Activation of ABA-responsive gene expression. Under osmotic stress conditions, the interaction with ABA leads to conformational changes in the ABA receptors [PYR (pyrabactin resistance)/PYL (PYR-related)/RCAR (regulatory component of the ABA receptor)], allowing them to interact with PP2Cs. PP2Cs act as a coreceptor to capture ABA, thereby suppressing its phosphatase activity. This sequestrates PP2Cs from SnRK2s, and free SnRK2s phosphorylate the downstream transcription factors AREB/ABFs. The phosphorylated AREB/ABFs trigger the transcription of numerous ABA-responsive genes.

**Figure 2 ijms-21-09517-f002:**
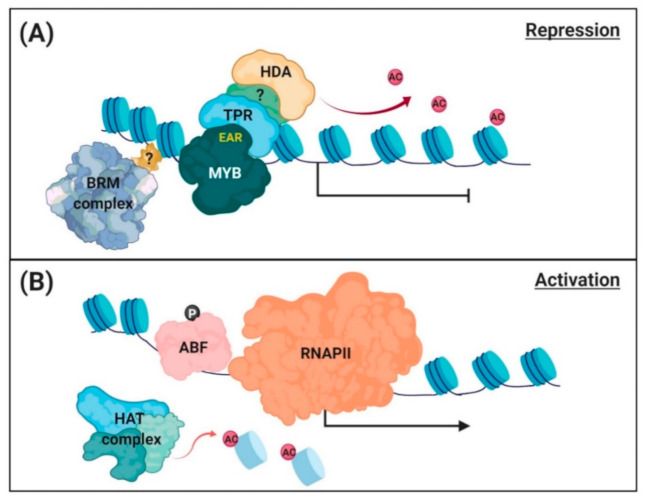
A working model of chromatin remodeling for regulation of PP2C gene transcription. (**A**) Repression of PP2C gene transcription. Under normal conditions, enhancer of ABA coreceptor (EAR) motif-containing MYB repressors (AtMYB44 and AtMYB20) interact with a TOPLESS-related corepressor (TPR), which recruits histone deacetylase (HDA) to suppress PP2C gene transcription. The chromatin remodeler, BRM-containing SWI/SNF complex, occupies the promoter and contributes to the repression of PP2C gene transcription. (**B**) Activation of PP2C gene transcription. Under osmotic stress conditions, the repressor is released from the promoter, and histone acetyltransferases (HATs) that acetylate the histones and relax DNA–histone binding in chromatin. Activator (AREB/ABFs) binds to the open promoter region, and RNA polymerase II (RNAPII) accesses and starts gene transcription.

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
