# Peer review of "Transcriptional Regulation of Protein Phosphatase 2C Genes to Modulate Abscisic Acid Signaling"

_ijms, 2020, doi:10.3390/ijms21249517_

Round 1

Reviewer 1 Report

The Review by Jung et al. is devoted to analysis of signal pathways of ABA. The work is interesting; however, there are questions and remarks:

   1. Introduction includes only briefly information about influence of ABA on physiological processes. However, it is known that ABA signaling interacts with processes of photosynthesis, transport of ions, electrical signaling, etc. ABA influence on physiological processes in plants can be related to Ca2+ influx, H2O2 production, decrease of activity of H+-ATP-ase, etc. Additionally, ABA signaling can participate in responses on heating, salinity, crushing, etc. Thus, I suppose that Introduction should be extended. ABA significance for physiological processes should be analyzed in more details.

   2. P. 3: Figure has not legend. It should be corrected.

   3. P. 3, lines 123-125: What ways of ABA transport into cell? If ABA interacts with intracellular receptors, the way should be discussed.

   4. P. 3-4, lines 125-127: Complex (complexes?) of ABA with these receptors should be described in more details.

   5. P. 4, lines 134: What are PYR1, PYL1, PYL2, PYL4, PYL5 and PYL8 differed? It should be clarified.

   6. P. 4, lines 143-144: “The ABI1–PYR1 interaction was stimulated within 5 min of ABA treatment in Arabidopsis [72]”. It is very fast effect. I suppose that method of treatment (which was used in [72]) should be described in more details.

   7. P. 4, line 175: What are SnRK2.6 and OST1 differed?

   8. P. 7: Figure has not legend. It should be corrected.

   9.  Additionally, problem of interaction between ABA signal and other pathways of stress signaling (jasmonate signaling, electrical signaling, etc) is very interesting. I suppose it should be discussed in the Review.

    Thus, I suppose that revision is necessary.

Author Response

Answers to Reviewers’ comments

Open Review 1

  1. Introduction includes only briefly information about influence of ABA on physiological processes. However, it is known that ABA signaling interacts with processes of photosynthesis, transport of ions, electrical signaling, etc. ABA influence on physiological processes in plants can be related to Ca2+influx, H2O2production, decrease of activity of H+-ATP-ase, etc. Additionally, ABA signaling can participate in responses on heating, salinity, crushing, etc. Thus, I suppose that Introduction should be extended. ABA significance for physiological processes should be analyzed in more details.

We expanded our description of the significance of ABA for physiological processes in the Introduction, as follows: [Lines 74-81]

        ABA also plays pivotal roles in various physiological processes during the plant life cycle, including seed dormancy, germination, lateral root formation, light signaling convergence, and control of flowering time [5,7,12]. These functions of ABA are related to Ca2+ influx, the production of reactive oxygen species such as H2O2, ion transport, and electrical signaling [11,12,27]. During these processes, ABA signaling interacts antagonistically or synergistically with other hormonal signaling pathways mediated by auxin, cytokinin, ethylene, and jasmonates [7]. Thus, excess ABA impairs developmental processes such as senescence, as well as pollen fertility, and also leads to seed dormancy and susceptibility to diseases [28].

  1. P. 3: Figure has not legend. It should be corrected.

The Figure 1 caption is now beneath the figure. [Lines 118-129]

  1. P. 3, lines 123-125: What ways of ABA transport into cell? If ABA interacts with intracellular receptors, the way should be discussed.

We have revised our explanation of ABA transport, as follows: [Lines 140-144]

                  ABA molecules biosynthesized in vascular tissues are distantly transmitted to sites such as guard cells to activate the closure of stomata [13, 14]. Multiple ABA transporters have been identified in Arabidopsis, including exporters (AtABCG25 and AtDTX50) and importers (AtABCG40 and AtAIT1) [51–55]. Guard cells themselves also biosynthesize ABA, which is sufficient for stomatal closure in response to low air humidity [56].

  1. P. 3-4, lines 125-127: Complex (complexes?) of ABA with these receptors should be described in more details.

We have revised our description of the receptor–ABA–PP2C complex, as follows. [Lines145-151]

                  ABA molecules are perceived intracellularly by soluble receptors predominantly located in the nucleus and cytosol of guard cells [16, 57]. A number of synonymous ABA receptors, e.g., pyrabactin resistance (PYR), PYR-related (PYL), and regulatory component of the ABA receptor (RCAR), have been identified as PP2C-interacting proteins in Arabidopsis [58–60]. PP2Cs have direct physical interactions with ABA and ABA receptors; these interactions are required for high-affinity binding of ABA [61,62]. Each PP2C functions as an ABA co-receptor within a holoreceptor complex that is constructed in combination with a particular PYR/PYL/RCAR.

  1. P. 4, lines 134: What are PYR1, PYL1, PYL2, PYL4, PYL5 and PYL8 differed? It should be clarified.

We described the functional differences among receptor members as follows: [Lines 150-151, 161-165]

                  Each PP2C functions as an ABA co-receptor within a holoreceptor complex that is constructed in combination with a particular PYR/PYL/RCAR.

                  Dittrich et al. [67] proposed that response specificity is achieved when the signals stimulate different members of the PYR/PYL/RCAR receptor family; PYL2 is sufficient for ABA-induced guard cell responses, whereas PYL4 and PYL5 are essential for the responses to CO2. Different combinations of PYRs and PP2Cs influence ABA binding affinity, and therefore affect the ABA sensitivity of the whole plant [68,69].

  1. P. 4, lines 143-144: “The ABI1–PYR1 interaction was stimulated within 5 min of ABA treatment in Arabidopsis[72]”. It is very fast effect. I suppose that method of treatment (which was used in [72]) should be described in more details.

We rewrote this sentence as follows: [Lines 168-171]

                  Nishimura et al. [74] performed co-immunoprecipitation experiments in a transgenic Arabidopsis line stably transformed with yellow fluorescent protein (YFP)–ABI1 fusion genes using a PYR1 antibody, and observed that the ABI1–PYR1 interaction was induced within 5 min after exogenous ABA application.

  1. P. 4, line 175: What are SnRK2.6 and OST1 differed?

These represent the same protein. Open Stomata 1 (OST1) is a synonym of SnRK2.6.

  1. P. 7: Figure has not legend. It should be corrected.

The Figure 2 caption is now beneath the figure. [Lines 301-310]

  1. Additionally, problem of interaction between ABA signal and other pathways of stress signaling (jasmonate signaling, electrical signaling, etc) is very interesting. I suppose it should be discussed in the Review.

In this article, we reviewed how plants modulate the ABA signaling pathway, focusing on the transcriptional regulation of PP2C gene expression by ABA. We have added the following two sentences in the revised manuscript: [Lines 77-79, 103-104]

                  During these processes, ABA signaling interacts antagonistically or synergistically with other hormonal signaling pathways mediated by auxin, cytokinin, ethylene, and jasmonates [7].

                  Kumar et al. [12] reviewed the integration of ABA signaling with other signaling pathways in development and plant stress responses.

Reviewer 2 Report

The authors reviewed the ABA signaling focusing on the regulation of PP2C genes.

Here are my comments:

  1. I would suggest moving the part from line 89 to line 104 to the introduction, for example after line 61. Having a brief description of the PP2C in the introduction will help the text flow.
  2. I think that the part from line 224 to 232 needs to be rephrased. They wrote: “AtMYB44 acts as a negative regulator of ABA signaling and stress responses, in contrast to its positive role in ABA responses indirectly suppressing PP2C gene transcription”, but also PP2C is part of the ABA signaling. I don’t understand what the authors consider ABA signaling and what ABA response.
  3. In line 242 authors wrote: “that acetylate histones to evict nucleosomes”. I am not sure that the main function of histone acetylation is nucleosome eviction. I would say that histone acetylation relaxes the DNA-histone binding.
  4. In the last paragraph “4.3. Osmotic stress memory”, the authors don’t mention the role of DNA methylation, which has been extensively studied.
  5. Sometimes when genes are mentioned their names are not in Italic.

Author Response

Answers to Reviewers’ comments

Open Review 2

  1. I would suggest moving the part from line 89 to line 104 to the introduction, for example after line 61. Having a brief description of the PP2C in the introduction will help the text flow.

As suggested, this section was moved to the Introduction. [Lines 51-65]

  1. I think that the part from line 224 to 232 needs to be rephrased. They wrote: “AtMYB44 acts as a negative regulator of ABA signaling and stress responses, in contrast to its positive role in ABA responses indirectly suppressing PP2Cgene transcription”, but also PP2C is part of the ABA signaling. I don’t understand what the authors consider ABA signaling and what ABA response.

This sentence has been rephrased as follows: [Lines 247-249]

                  These results suggest that AtMYB44 may act as a negative regulator of ABA signaling, which is inconsistent with its reported indirect positive role of suppressing PP2C gene transcription.

  1. In line 242 authors wrote: “that acetylate histones to evict nucleosomes”. I am not sure that the main function of histone acetylation is nucleosome eviction. I would say that histone acetylation relaxes the DNA-histone binding.

This sentence has been rephrased as follows. [Lines 259-261]

                  Activators loaded on the promoter recruit co-activators and histone acetyltransferases (HATs) that acetylate the histones and relax DNA–histone binding in chromatin [112].

  1. In the last paragraph “4.3. Osmotic stress memory”, the authors don’t mention the role of DNA methylation, which has been extensively studied.

We have added a sentence to the last paragraph mentioning the extensively studied role of DNA methylation, as follows: [Lines 382-386]

                  Induced changes in the DNA methylation state were suggested as a possible mechanism by Zheng et al. [161], who observed that multi-generational drought stimulation induced the non-random appearance of epimutations and inheritance of high methylation state in advanced rice plant generations.

  1. Sometimes when genes are mentioned their names are not in Italic.

We found and corrected a few mistakes, as follows: [Lines 154, 233]

                  PYR1      PP2CA

Round 2

Reviewer 1 Report

Authors essentially improved the manuscript. I do not have any comments or questions.